# Diseases Associated with Defects in tRNA CCA Addition

**DOI:** 10.3390/ijms21113780

**Published:** 2020-05-27

**Authors:** Angelo Slade, Ribal Kattini, Chloe Campbell, Martin Holcik

**Affiliations:** Department of Health Sciences, Carleton University, Ottawa, ON K1S 5B6, Canada; ziggyslade@cmail.carleton.ca (A.S.); ribalkattini@cmail.carleton.ca (R.K.); chloecampbell@cmail.carleton.ca (C.C.)

**Keywords:** tRNA, TRNT1, CCA, sideroblastic anemia, mitochondrial disease, tRNA nucleotidyl transferase 1

## Abstract

tRNA nucleotidyl transferase 1 (TRNT1) is an essential enzyme catalyzing the addition of terminal cytosine-cytosine-adenosine (CCA) trinucleotides to all mature tRNAs, which is necessary for aminoacylation. It was recently discovered that partial loss-of-function mutations in TRNT1 are associated with various, seemingly unrelated human diseases including sideroblastic anemia with B-cell immunodeficiency, periodic fevers and developmental delay (SIFD), retinitis pigmentosa with erythrocyte microcytosis, and progressive B-cell immunodeficiency. In addition, even within the same disease, the severity and range of the symptoms vary greatly, suggesting a broad, pleiotropic impact of imparting TRNT1 function on diverse cellular systems. Here, we describe the current state of knowledge of the TRNT1 function and the phenotypes associated with mutations in *TRNT1*.

## 1. Introduction

Mitochondrial diseases are a large group of clinically and genetically heterogeneous disorders. Clinically, they present at any age, show a wide spectrum of organ system dysfunction, and range in severity from relatively mild disease to neonatal or even antenatal lethal presentations. Prevalence estimates are variable but, conservatively, mitochondrial diseases affect at least 1 in 8500 persons [1]. However, mitochondrial dysfunction has been linked to the pathology of chronic diseases as well, including cancer, type 2 diabetes, and Alzheimer’s disease, among others [2]. Research into mitochondrial disease processes and treatments is limited, yet has the potential to greatly improve care and outcomes for affected individuals and is also likely to have implications for those affected by more common chronic diseases that have features of mitochondrial dysfunction (e.g., Alzheimer’s disease, Parkinson’s disease, and diabetes) [3]. Genetically, disease-causing mutations have been described in genes on both the mitochondrial (mt) and nuclear genomes. There have been 13 identified proteins encoded by mitochondrial genes, which have been linked to mitochondrial disorders [4]. Mitochondrial genes also encode 22 tRNAs and 2 rRNAs. However, many components of oxidative phosphorylation, including complexes I–IV, ATP synthase, and mitochondrial DNA (mtDNA) replication and transcription factors are encoded in the nuclear genome [5].

Commonly, defective oxidative phosphorylation (OXPHOS) is seen in mitochondrial disease, but heterogeneous clinical presentations can be seen with mutations both in mitochondrial and nuclear genomes [5]. A common nuclear-encoded cause of mitochondrial disease are mutations in genes involved in mitochondrial protein synthesis. These mutations often result in tissue-specific effects. Phenotypes associated with nuclear mutations are often severe and early onset, and many times fatal. Nuclear genetic mutation may also have a secondary effect of causing instability in the mitochondrial genome, and these mitochondrial diseases can therefore be inherited in a fashion consistent with Mendelian genetics. Instability can be caused by depletion of mtDNA, point mutations, or deletions [5].

Strikingly, the majority of mt DNA mutations are found in tRNA genes, and many of the mutations in nuclear genes are associated with nucleic acid metabolism, including tRNA maturation [6]. The molecular mechanism(s) by which these mutations cause such a broad range of disorders are not fully understood. It is believed, however, that in addition to affecting the canonical functions of tRNAs (primarily in mitochondria protein synthesis), the disease-causing mutations impact non-canonical functions as well (e.g., as a signaling molecule, in the cellular stress response), adding to the complexity of mitochondrial diseases.

## 2. TRNT1 and tRNA Maturation

Maturation of tRNAs (both mitochondrial and cytoplasmic) is a multi-step process that requires 5′ and 3′ processing, extensive base modifications, CCA addition, and aminoacylation. For cytoplasmic tRNAs, these steps occur in a precise order and are performed in distinct cellular compartments; in contrast, all mt tRNA processing steps occur in mitochondria [7]. The CCA addition is performed by a unique CCA-adding enzyme, tRNA nucleotidyltransferase (TRNT1 in human, or CCA1 in yeast; EC 2.7.7.72), which is located on the third human chromosome at position 26.2 and has 7 exons with a span of approximately 20kb [8]. *TRNT1* encodes the only human CCA-adding enzyme, an RNA polymerase required for the post-transcriptional, template-independent addition of two cytosines and one adenosine to the 3′ end of both cytosolic and mitochondrial tRNAs [8]. This 3′ addition is required for aminoacylation, correct positioning on the ribosome, and subsequent protein synthesis [9]. Following the addition, the CCA trinucleotide acts as an anti-determinant for 3′ endoribonuclease activity [10]. In the mitochondria, the addition of CCA (and subsequent 3′ protection) has been proposed to stabilize select tRNA in order to achieve the balanced tRNA levels observed in the mitochondria [11]. The 3′ CCA tRNA terminus is subject to frequent cleavage or degradation and TRNT1 also functions to maintain and repair previously added CCA sequence [12]. Importantly, TRNT1 can discriminate against tRNA backbone damage, keeping damaged tRNAs from incorporation into the translation machinery (see below) [13]. Of note, TRNT1 appears to play additional, non-canonical roles in the cell. These include assisting in tRNA nuclear export and quality control as a nuclear export adaptor protein [14], assessment of tRNA structure and targeting of unstable tRNA for decay [15,16], and addition of CCA to small RNAs [16,17].

### 2.1. TRNT1 Structure and Mechanism of Action

The primary role of TRNT1 in mammalian cells is the addition of CCA trinucleotide to tRNA termini. TRNT1 is classified in full as a template-independent class II RNA nucleotidyltransferase enzyme. Although similar in function to each other, the class II CCA-adding enzymes differ in sequence significantly from the class I enzymes. In addition, class I and class II enzymes are found in different life kingdoms. Class I are present in archaeal organisms, while class II are found in eukaryotes [18]. Although the sequence differences inform the specific mechanism of CCA addition, there is quite a bit of homology between the active sites of class I and class II enzymes hinting at common basic principles of CCA addition. Furthermore, it has been found that in some bacteria, the CCA addition is carried out by two separate enzymes—one which is responsible for adding “CC” and another one for adding an “A”. The flexibility between the head and neck domain of the CCA-adding enzyme is what allows for the CCA trinucleotide addition to be completed by a single enzyme (see below) [18]. Class I and class II CCA-adding enzymes share similarities to other polymerase enzymes, specifically within their active sites. However, CCA-adding enzymes are unique in that they are a template-independent polymerase. This means that they do not add the CCA end to tRNA based off a preexisting sequence encoded by the genome, rather the formation of the enzyme and its amino acid sequence itself select for the specific nucleotide bases (C, C, or A) to be added in a specific and precise sequence [9]. For the purpose of this review, we will focus on TRNT1 only. It should be noted, however, that the specific mechanistic details of class I enzymes function are more robustly understood than those of their eukaryotic counterparts [18].

TRNT1 protein can be divided into four distinct functional domains: head, neck, body, and tail (Figure 1) [18]. Each of these domains fulfills a specific function within TRNT1; however, the two most critical domains are the head and neck, as these are the domains which house and assist in the movement of the active site for CCA addition. The enzyme active site is comprised of two aspartic acid residues, Asp48 and Asp50, and is located within the head domain (Figure 2) [13]. There are two other important residues in the head domain—glutamic acid, Glu164, and arginine, Arg168—which, due to the flexible properties of their side chains, allows the binding pocket of the active site to discriminate between cytosines and adenines, and incorporate the correct nucleotide at each step of CCA addition. The second most important domain of TRNT1 protein is the neck domain, which interacts with the head domain to give the enzyme its flexibility. Between the head and neck domains, there is a positively charged region of residues where the tRNA itself binds to TRNT1. The 5′ end of the acceptor stem is the part of the tRNA which is tightly bound to TRNT1, while the 3′ end is more loosely attached [19]. This interesting aspect of the RNA–protein complex formation is what allows for the tRNA to be tightly held to TRNT1, while at the same time allowing the 3′ end to move and be freely added to. The body and tail regions of the TRNT1 protein are not important for the CCA-adding function of the active site, but are responsible for identifying the structural components which define the tRNA, such as acceptor stem structure [18].

The mechanistic steps of CCA addition by TRNT1 are briefly outlined in Figure 1A. When TRNT1 encounters a tRNA molecule lacking a CCA 3′ end, or with a damaged 3′ end, the enzyme will form a RNA–protein complex with the tRNA, binding tightly to the 5′ end of the acceptor stem [18]. Through this attachment, TRNT1 adopts its first conformational state, which prompts the addition of the first cytosine to the 3′ end of the tRNA. The addition of the first C to the 3′ end causes the head–neck joint to flex and positions the active site into a conformation which favors the addition of the second C. After both cytosines have been added, the RNA–protein complex adopts its final conformation, which allows for the adenosine to be added. With the complete CCA terminus now constructed, the RNA–protein complex dissociates and the tRNA is released. The release of tRNA allows for TRNT1 to return to its original conformation, and to engage another tRNA molecule. It should be noted that this is the current model for CCA addition by class II CCA-adding enzymes; however, the precise molecular steps are still not fully understood.

### 2.2. TRNT1 and tRNA Quality Control

Although TRNT1 is primarily known for its role in CCA addition, it has also been shown to play a role in tRNA quality control. All organisms possess a functional homolog of TRNT1, be it a class I or class II tRNA nucleotidyltransferase enzyme. Interestingly, however, not all organisms require a post-transcriptional CCA-adding function. For example, in *Escherichia coli*, the 3′ CCA end is already encoded within the DNA genes for tRNAs, and the correct terminal CCA is added during transcription [13]. Yet, *E. coli* still possesses a CCA-adding enzyme [13]. It is believed that the tRNA nucleotidyltransferase enzyme is still required, however, due to its ability to recognize and repair CCA ends, which are vulnerable to damage during the tRNAs life in the cell [13]. The 3′ CCA ends of all tRNAs are essential for the proper function of the aminoacyl-tRNA synthetases (aaRS), a class of enzymes, which are responsible for the addition of the amino acids to their cognate tRNAs [18]. The damage to CCA termini can result in improper aminoacylation, where the incorrect amino acids could be attached to the wrong tRNAs [19]. Consequently, such mis-aminoacylation could result in mutations in proteins at the level of translation due to incorrect amino acids being incorporated into the proteins. TRNT1 can detect and repair partially damaged CCA termini [18].

Figure 1B outlines an additional mechanism of tRNA quality control performed by TRNT1, one that detects structurally unstable tRNAs and tags them for degradation. The structural quality of tRNAs is paramount for their ability to deliver appropriate amino acids to translation apparatus. Unstable tRNAs may arise due to loss of base modifications that stabilize secondary tRNA structure, or due to point mutations disrupting base pairing [21]. During the CCA addition cycle to a tRNA, the TRNT1 enzyme exerts pressure onto the tRNA, which creates tension in the tRNA. If the tRNA is structurally sound, then this tension does not affect the structure of the tRNA, the RNA–protein complex dissociates properly to release the modified tRNA, and TRNT1 resets to its natural position ready to engage another tRNA [19]. However, if the tRNA is structurally unstable, it will “buckle” under the pressure and will undergo additional modification by TRNT1. Under the pressure of TRNT1, the tRNA will “slip” backward on its 3′ end and form a temporary bubble upstream of the 3′ end. This results in the CC of the CCA end binding with their cognate nucleotides (G) on the 5′ end. This slippage and formation of the temporary nucleotide bubble allows the tRNA to appear to TRNT1 as if it does not currently have a CCA end attached. Due to this, TRNT1 will revert to its default conformation and begins a second round of CCA addition. On the completion of this second CCA addition, the RNA–protein complex is dissociated and the released tRNA is stretched back out so that the CCA end is free and not bound to any nucleotides on the 5′ end of the tRNA. This newly formed CCACCA sequence functions as a signal to 3′–5′ exonucleases that target the tRNA for degradation [19]. Therefore, through the process of the second CCA addition TRNT1 can identify structurally unstable tRNAs, and mark them for degradation to prevent these aberrant molecules from potentially negatively impacting protein synthesis. It is important to note that TRNT1 can only recognize and tag for degradation structurally unstable tRNA that contain GG sequence at the 5′ end of the acceptor stem, which is present in approximately 45% of all tRNAs [15]. For all other tRNAs, a CCACCA-independent mechanism executed by poly(A) polymerases is used [19].

## 3. Mitochondria and Disease

The mitochondria are an essential and complex organelle, with roles ranging from ATP production via oxidative phosphorylation, to regulation of apoptosis, to homeostasis of ions [2]. Thus, dysfunction of the mitochondria can have serious, and sometimes fatal results on organ systems and their functioning. Mitochondrial diseases are a large group of heterogeneous disorders caused by dysfunctional mitochondria. The disease-causing mutations have been described in genes on both the mitochondrial and nuclear genomes; most commonly (80%–85%) resulting from defects in nuclear encoded genes [22,23]. The majority of pathogenic mtDNA mutations have been described in genes encoding tRNA, while nuclear DNA mutations have been found in genes encoding a wide variety of proteins targeted to the mitochondria, including subunits of the respiratory chain complex and their assembly, enzymes involved in mtDNA replication and maintenance of the nucleotide pools, mitochondrial protein translation and mitochondrial protein import. Mitochondrial diseases generally show phenotypes in tissues with high-energy requirements (e.g., muscle, neurons, endocrine, heart), and clinical presentations typically involve these tissues (e.g., skeletal myopathy, cardiomyopathy, seizures, peripheral neuropathy, deafness, bone marrow dysfunction, etc.) [24]. In addition, diseases of aging such as type 2 diabetes, Parkinson’s and Alzheimer’s disease, atherosclerotic heart disease, stroke, and many forms of cancer, have been associated with defects in mitochondrial function [25]. Although this suggests that the disease pathology is caused by a deficiency in mitochondrial energy metabolism, it has been difficult to rule out that defects in the mitochondria impinge on other cellular pathways, thereby causing disease. Furthermore, it has been estimated that 15% of proteins that function in mitochondria also function outside of mitochondria, with approximately one-third of cytoplasmic proteins that function in the mitochondria being co-localized to additional cellular compartments [26,27].

Interestingly, a large proportion of mitochondrial diseases are caused by defects in genes associated with tRNA metabolism [6]. While tRNA genes only comprise approximately 10% of the mitochondrial genome, of the more than 150 mutations in the mitochondrial genome associated with human disease, more than 60% are associated with these genes [28]. Although mitochondria encode their own set of transfer tRNAs (mt tRNA), the proteins necessary for processing and maturation of mt tRNAs are encoded in the nucleus and subsequently imported into the mitochondria [29]. The fact that the same enzymes also modify cytoplasmic tRNAs may add to the large clinical heterogeneity of mitochondrial diseases. Mutations in these proteins affect aminoacylation, abundance, expression, modifications and maturation of tRNAs, thereby impinging on the translation of mitochondrial genes. However, it has also been recognized that some of the pathogenic mutations may affect non-canonical functions of tRNAs, thus contributing to disease without impacting translation [29].

## 4. TRNT1 and Disease

Although the CCA addition has been studied extensively in prokaryotes and lower eukaryotes for over thirty years [30], the connection between TRNT1 and human disease has been made only recently. Early genetic mapping of the candidate genes associated with non-syndromic mental retardation on chromosome 3p (MRT2A) identified a region containing nine genes including *TRNT1* [31]. Although all the affected individuals contained a homozygous G→A base pair change in a 3′–splice site of *TRNT1*, 30% of the normal controls also harboured this mutation. *TRNT1* was also identified in a differential gene expression study comparing colorectal cancer patients of African American and European origins [32]. The TRNT1 was shown to be downregulated in the cohort of African American patients, suggesting a link with higher incidence and mortality of colorectal cancer seen in African Americans. However, the first direct link between *TRNT1* and disease was reported only in 2014. At that time, a young female patient presented with sensorineural hearing impairment, sideroblastic anemia, hyperalaninemia, episodic fever, B-cell immunodeficiency, and developmental delay (SIFD; OMIM #616084). SIFD is a recently identified severe syndromic form of congenital sideroblastic anemia [4]. A whole-exome sequencing strategy was therefore pursued to identify disease-causing mutations in this patient. Two non-conservative missense mutations in phylogenetically highly conserved residues were found in the *TRNT1*. In parallel, hypomorphic mutations in the same gene were identified in a cohort of additional 10 clinically similar patients using homozygosity mapping in nine pedigrees. Using biochemical, cell and yeast approaches, we have demonstrated that the patients’ mutations are indeed causative of SIFD [33]. This conclusion was subsequently confirmed by an independent group that additionally showed defective CCA addition to the non-canonical mitochondrial tRNA^Ser(AGY)^, but not other mitochondrial or cytoplasmic tRNAs, and reduced mitochondrial translation in patient samples [34]. In contrast, similar investigation of an independent patient by Wedatilake et al. [35] disclosed a partial lack of CCA in mt-tRNA^Cys^ and mt-tRNA^Leu(UUR)^, but not in tRNA^Ser(AGY)^. We have further shown that although patient fibroblasts do not appear to be defective in cellular morphology, TRNT1 localization, global translation, mitochondrial network architecture, or mitochondrial transmembrane potential, the patient-derived fibroblast cells harbouring mutated *TRNT1* are defective in their ability to properly form OXPHOS complexes and consume oxygen, defects that likely contribute to the observed mitochondrial phenotypes associated with mutant *TRNT1* [36].

Since the original association of *TRNT1* deficiency with SIFD, several studies were published describing additional patients harbouring disease-causing mutations in *TRNT1* (Table 1). Interestingly, both the clinical symptoms as well as the molecular features identified in these patients are variable and do not always align with the original reports of SIFD. In addition to the invariable SIFD clinical features (congenital sideroblastic anemia, B-cell immunodeficiency, recurring fevers and developmental delay), some SIFD patients also exhibit ataxia, severe variable sensorineural hearing loss, splenomegaly, retinitis pigmentosa, seizures and cardiomyopathy. The severe, multi-organ nature of the disease often results in death within the first decade of life, with bone marrow transplantation the only reported curable option in one reported case [4]. DeLuca and colleagues described two non-syndromic retinosa pigmentosa families harbouring disease-causing variants in *TRNT1* [37]. Interestingly, the affected individuals were otherwise healthy with no history of developmental delay, deafness, ataxia, seizures or cardiac disease, features normally associated with SIFD. However, they all exhibited low normal hemoglobin and hematocrit, microcytosis and an elevated erythrocyte distribution width. The molecular examination revealed decreased but detectable levels of TRNT1 protein in patient-derived fibroblasts. The authors used a zebrafish model to demonstrate the functional link between *TRNT1* deficiency and RP. Morpholino-mediate *TRNT1* knockdown recapitulated anemia and sensory organ defects that are seen in SIFD patients, as well as decreased CCA addition to tRNA. Importantly, titration of TRNT1 levels resulted in visual dysfunction only, and this phenotype was partially corrected by the expression of human TRNT1.

Hull et al. described a consanguineous family with three affected children that had poor growth, microcephaly, borderline microcytic, hypochromic anemia without sideroblasts, moderate pan-hypogammaglobulinemia without B-cell lymphopenia, but no developmental delay [32]. All three siblings displayed relatively severe generalized inner retinal dysfunction upon detailed electrophysiological testing; however, whole-exome sequencing did not identify mutations in known recessive cataract genes. In contrast, a homozygous mutation in TRNT1 (R99W) was identified in all three siblings. This mutation has not been previously associated with a disease phenotype. It lies in the N-terminal portion of the TRNT1 protein, away from the active site and is predicted to be damaging in silico. Although retinal abnormalities are a known association of mitochondrial disease [38], this report was the first to link *TRNT1* deficiency to an inner retinal dysfunction.

Recently, Frans et al. described an adult patient who exhibited B-cell immunodeficiency, bilateral congenital cataracts, dysmorphic features, mild psychomotor retardation, bilateral sensorineural hearing loss and inflammatory bowel disease [39]. Whole-exome sequencing identified a homozygous missense mutation in *TRNT1* (R99W). In patient-derived fibroblasts, the levels of TRNT1 appeared normal, excluding a possible impact on protein stability. The patient exhibited elevated levels of serum IFN-α protein and increased expression of interferon-stimulated genes. The authors noted that this signature could be indicative of type I interferonopathy; indeed some SIFD clinical features are shared with diseases characterized by an upregulation of type I interferon signaling (e.g., periodic fevers, cardiomyopathy, neurological abnormalities, developmental delay) [39].

Lougaris and Chou characterized a single patient harbouring two heterozygous mutations in *TRNT1* (I155T; R203K) [40]. In addition to the clinical phenotype consistent with SIFD this patient also presented with T cell abnormalities. This was also the first report to show a reduction in T cell receptor excision circles (TRECs) and kappa-deleting recombination excision circles (KRECs) in an SIFD patient. Given the increased availability of routine TREC measurement, the authors further propose to use this test as an early screening test for these clinically heterogeneous diseases.

Barton et al. described an unusual case of severe chronic antenatal anemia accompanied by extramedullary hematopoiesis, a high number of circulating nucleated red blood cells, and fetal hydrops resulting in an early neonatal death [41]. Bi-allelic mutations in *TRNT1* (c.608+1G>T and c.668T>C) were confirmed in the affected individual suggesting that *TRNT1* deficiency is the underlying cause of the disease. This report suggests that SIFD in its more severe form can present at birth with additional clinical features and that genetic testing of cases of fetal hydrops accompanied by severe anemia and extramedullary hematopoiesis might increase the frequency of reported *TRNT1* deficiency. Interestingly, the sibling of the index patient, carrying the identical bi-allelic mutations in *TRNT1*, did not present with hydrops, although they both shared prominent extramedullary hematopoiesis and a high number of circulating nucleated red blood cells.

Giannelou et al. studied nine additional patients with SIFD [42]. These patients presented with a spectrum of clinical features consistent with SIFD and the whole-exome or targeted sequencing identified pathogenic bi-allelic mutations in *TRNT1* in all of them. Three of these mutations were previously not associated with SIFD. Similarly to the DeLuca study [37], zebrafish model with complete loss of function mutations in *TRNT1* recapitulated several of the patient phenotypes. Detailed molecular analysis performed in patient-derived fibroblasts from select patients revealed decreased abundance of mature tRNAs and increased reactive oxygen species production in patients’ samples. In addition, the authors observed increased levels of pro-inflammatory cytokines in patient serum samples, when compared to normal controls. Importantly, several patients were treated with TNF inhibitors and showed clinical improvement over an extended period of time. The improvement included decreased frequency of febrile illness, fewer blood transfusions and normalization of inflammatory biomarkers.

In the previously mentioned paper by Wedatilake et al. [35], another unique case is reported in which a patient experienced severe cerebral atrophy during the progressed of the disease towards the end of life. Lesions and abnormalities began to develop within the brain, which were described by the authors as bearing similarity to that of an infection within the brain. Initially, an MRIs showed no evidence of any cerebral atrophy, but as the disease progressed this became an issue.

Bader-Meunier et al. reported four cases of early-onset panniculitis associated with inherited B- or T cell immunodeficiency [43]. Of the four cases reported, compound heterozygous mutation in *TRNT1* (c.1213G>A/c.1057-7C>G) was identified in one child, expanding the repertoire of phenotypes associated with *TRNT1*. In addition to panniculitis, this patient also exhibited periodic febrile illness, chronic microcytosis and B cell deficiency, but no other clinical features typically seen in SIFD.

## 5. Conclusions

The inheritance pattern of TRTN1-associated diseases is autosomal recessive and the number of cases is too low to estimate data on the prevalence of various mutations, correlations with specific ethnic background or familial histories of consanguinity. The wide range of symptoms associated with the different mutations within *TRNT1* as well as its rarity within the population prevents development of routine genetic screening for *TRNT1* mutations. In addition, due to the often short lifespan of patients, no meaningful distinction has been drawn between pediatric and adult presentation of the disease. As seen from the above summary of reported cases, not all *TRNT1* deficiency associated symptoms appear to be related to metabolic syndrome, or dysfunctional mitochondria. The severity of the presented symptoms, as well as the life expectancy of the patients, is also highly variable. The precise reasons for these differences are not known, but are likely correlated with the location of mutations in *TRNT1*, and whether they occur as homozygous or compound heterozygous (Figure 2 and Figure 3). Mapping of the location of pathogenic variants on *TRNT1* suggests that milder forms of the disease are associated with mutations found in the N- or C-terminal portions of the protein, whereas the more severe phenotype is associated with mutations in the central, catalytic part of the protein. Indeed, a recent study investigating the biochemical properties of mutant TRNT1 proteins shows that while some mutations (e.g., T154I, L166S) impair the protein stability, other mutations (e.g., M158V, R190I and I223T) impact both the CCA-adding catalytic activity as well as protein stability [44]. In addition, since TRNT1 is encoded in the nucleus but required both in the cytoplasm and in mitochondria, specific mutations may impact the ability of TRNT1 to refold properly once transported into mitochondria, thus giving rise to the spectrum of observed phenotypes associated with *TRNT1* deficiency [44,45]. However, the full understanding of the genotype–phenotype relationship associated with *TRNT1* mutations, and the tissue specificity of many clinical features of *TRNT1* patients, will require systematic and in-depth molecular investigation.

Given the complexity and broad range of the *TRNT1*-associated pathologies, the current treatments are focused on attempting to improve the patient’s quality of life. Bone marrow transplant has been attempted in few cases. However, as briefly discussed above, one surviving patient who received bone marrow transplant was cured of the immunological part of SIFD (B-cell immunodeficiency). This currently remains the only completely successful attempt at this semi-curative treatment.

## Figures and Tables

**Figure 1 ijms-21-03780-f001:**
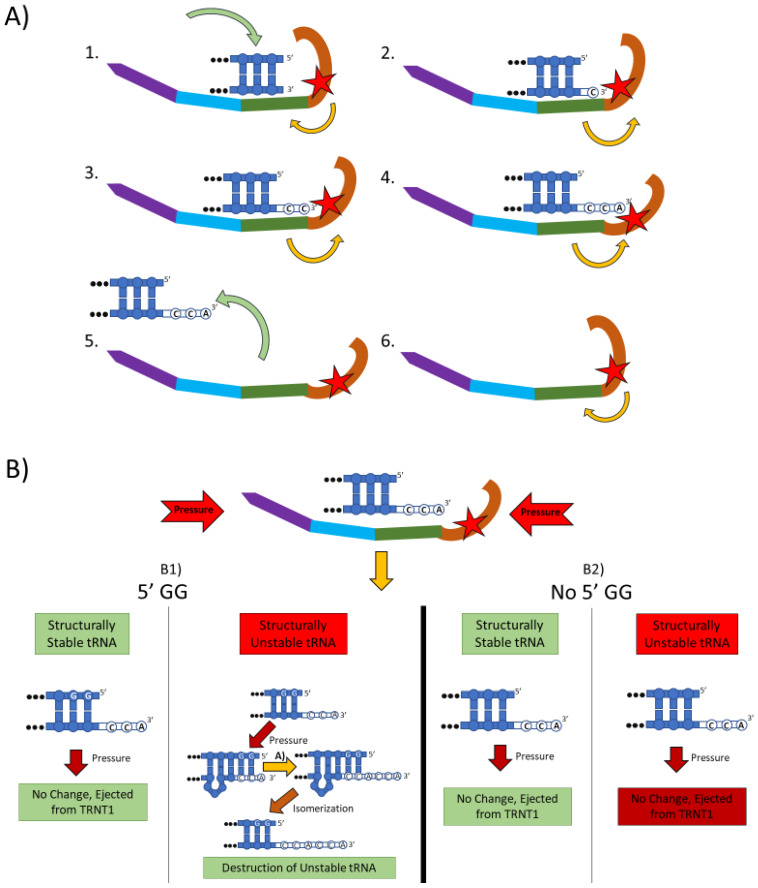
The two main functions of the TRNT1 enzyme: CCA addition, and tRNA quality control through CCA addition. (**A**) CCA addition. TRNT1 enzyme pictured with domains colored as: head (orange), neck (green), body (blue), and tail (purple). Additionally, the catalytic site is visualized by the red star, with the white nucleotides representing the end of the acceptor stem of the attached tRNA, and the yellow arrows indicating movement between the head and neck domains. (A1) tRNA binds with TRNT1 forming an RNA–protein complex. (A2) Cytosine is incorporated in a template-independent manner to the 3′ end of the tRNA, resulting in a conformational change in the TRNT1 enzyme. (A3) A second cytosine is added to the growing 3′ end of the tRNA, resulting in an additional structural shift in TRNT1. (A4) The terminal adenine is added to finish the CCA addition, and the structural integrity of the tRNA is judged. (A5) The RNA–protein complex dissociates and the tRNA is ejected from TRNT1. (A6) TRNT1 resets to its resting conformation. (**B**) tRNA quality control. During the addition of the CCA end to the tRNA, the TRNT1 exerts pressure on the tRNA, which can result in one of four outcomes. (B1) If the tRNA has two guanines on its 5′ end and the tRNA is structurally unstable, the pressure from TRNT1 will force the recently created CCA end to slide back and create a bubble of nucleotides. This allows for the addition of a second CCA end, tagging the tRNA for destruction. If the two 5′ guanines are present and the tRNA is structurally sound, then no bubble will form and no second CCA will be added. (B2) If there is no GG present on the 5-prime end of the tRNA, when the pressure is exerted from the TRNT1, no bubble is formed and the tRNA is ejected from the RNA–protein complex. This will occur whether the tRNA is structurally stable or not.

**Figure 2 ijms-21-03780-f002:**
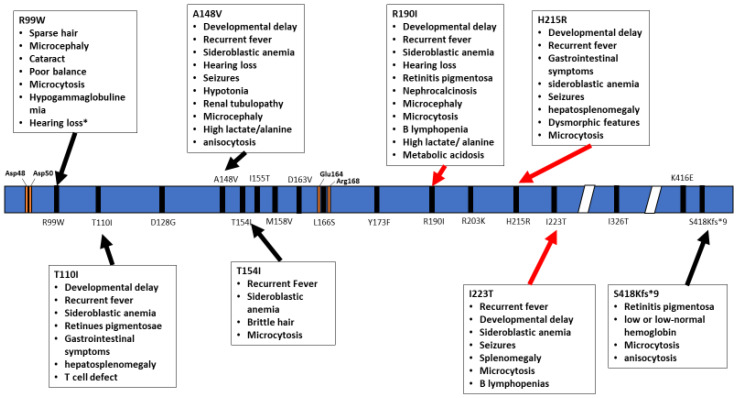
Schematic representation of the TRNT1 protein, with indicated positions of homozygous mutations in TRNT1. The specific phenotypes associated with these mutations are listed in boxed above and below the protein. Red arrows indicate severe symptoms while black arrows indicate mild phenotype. Orange marks indicate the active site as well as important residues to active site function. * Hearing loss reported for one patient harbouring the R99W mutation was suggested to occur due to a deletion in another gene, GJB2 [20].

**Figure 3 ijms-21-03780-f003:**
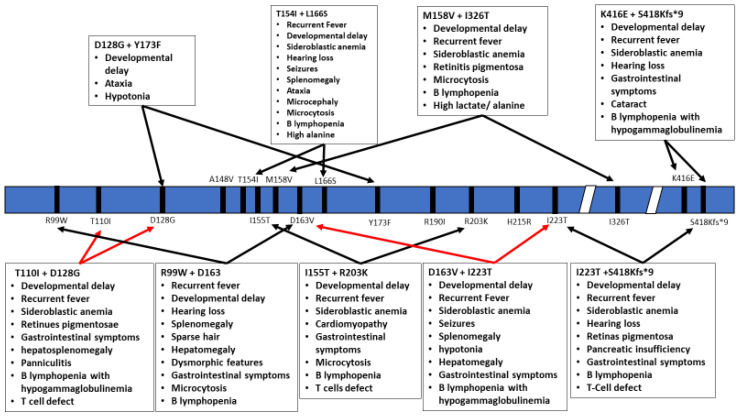
Schematic representation of the TRNT1 protein with indicated positions of heterozygous mutations in TRNT1. The specific phenotypes associated with these mutations are listed in boxes above and below the protein. Red arrows indicate severe symptoms, while black arrows indicate mild phenotype.

**Table 1 ijms-21-03780-t001:** Reported clinical and laboratory features and their frequencies associated with TRNT1 deficiency.

Clinical Features		Cases (Total Number)	%	Reference						
				[20]	[35]	[39]	[40]	[41]	[42]	[43]
General and administrative site disorder		28/35	80.00%							
	Febrile illness	26/35	74.29%	1/3	14/18	0/1	1/1	0/2	9/9	1/1
	Developmental delay	25/34	73.53%	0/3	14/18	1/1 (mild)	1/1	1/2 (mild)	8/9	
Blood and lymphatic system disorder		23/35	65.71%							
	Sideroblastic anemia	23/35	65.71%	0/3	13/18	0/1	1/1	1/2	7/9	1/1
	Splenomegaly	10/29	34.48%		4/18			1/2 (severe)	6/9	
Gastrointestinal disorders		19/29	65.52%							
	Inflammatory bowel disease	1/1	100.00%			1/1				
	Diarrhoea (is this IBD?)	15/28	53.57%		8/18	1/1			7/9	
	Vomiting	11/27	40.74%		7/18				4/9	
	Pancreatic insufficiency	3/18	16.67%		3/18				2/9	
	Gastrointestinal symptoms	10/10	100.00%				1/1		9/9	
Nervous system disorder		14/31	45.16%							
	Sensorineural hearing loss	13/31	41.94%	1/3	7/18	1/1			4/9	
	Seizures	9/27	33.33%		7/18				2/9	
	Ataxia	5/18	27.78%		5/18					
	Hypotonia	6/27	22.22%		5/18				1/9	
	Acute encephalopathy	2/18	11.11%		2/18					
	Poor balance	5/12	41.67%	2/3					1/9	
Congenital, familial and genetic disorders		15/33	51.51%							
	Microcephaly	3/3	100.00%	3/3						
	Villous atrophy	2/18	11.11%		2/18					
	Dysmorphic features	4/12	33.33%			1/1		0/2	3/9	
	Retinitis pigmentosa	10/27	37.04%		6/18				4/9	
	Congenital anemia of unknown cause	1/2	50.00%					1/2		
Eye disorder		6/13	46.15%							
	Cataracts	6/13	46.15%	3/3		1/1			2/9	
Skin and subcutaneous diseases		11/31	35.48%							
	Brittle hair	5/18	27.78%		5/18					
	Sparse hair	4/12	33.33%	3/3					2/9	
	Panniculitis	1/1	100.00%							1/1
Hepatobiliary disorders		10/29	34.48%							
	Hepatomegaly	10/29	34.48%		4/18			1/2	5/9	
Renal and urinary disorder		5/18	27.78%							
	Nephrocalcinosis	5/18	27.78%		5/18					
	Renal tubulopathy	4/18	22.22%		4/18					
Cardiac disorders		2/19	10.53%							
	Cardiomyopathy	2/19	10.53%		1/18		1/1			
Pregnancy, puerperium and perinatal conditions		1/2	50.00%							
	Fetal hydrops	1/2	50.00%					1/2		
Reproductive system and breast disorders		1/2	50.00%							
	Ovarian failure	1/2	50.00%	1/2 *						
Laboratory investigation										
Low or low-normal hemoglobin		16/18	88.89%		16/18					
Microcytosis		24/25	96.00%	1/1	16/18	3/3		1/1	2/9	
B lymphopenia with or without Hypogammaglobulinemia		22/31	70.97%	1/1	12/18			1/1	6/9	1/2
Hypogammaglobulinemia without B lymphopenia		3/3	100.00%			3/3				
Anisocytosis		9/18	50.00%		9/18					
High lactate		6/18	33.33%		6/18					
Metabolic acidosis		5/18	27.78%		5/18					
High alanine		3/18	16.67%		3/18					
Abnormal retinal ERG		3/3	100.00%			3/3				
T cell defects		3/10	30.00%					1/1	2/9	

* Note that there are only two females in this study and one male, and only one of the females has ovarian failure.

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
