# Peer review of "Diseases Associated with Defects in tRNA CCA Addition"

_ijms, 2020, doi:10.3390/ijms21113780_

Round 1

Reviewer 1 Report

This is a well written review which focuses on the diseases associated with partial loss-of-function mutations of tRNA nucleotidyl transferase 1 (TRNT1).

Could the authors please consider the following:

  • Although I enjoyed reading the Introduction and Section 2 (mitochondria and disease), I felt these two sections were far too long and it was only until Section 3 that the review started to discuss the focus of the paper. Could the authors combine Sections 1 and 2 to provide a more succinct introduction or rearrange the content so that the focus of the paper is introduced to the reader earlier on.
  • Could the text in Figure 1 be increased in size as the font is too small to read.
  • I would suggest including tRNA nucleotidyl transferase 1, tRNA CCA and TRNT1 as key words
  • Please include the Enzyme Commission number for TRNT1
  • Are there any other names that TRNT1 is known by?
  • Could the authors please provide information on the chromosome location of TRNT1, the number of exons and span size.
  • Could the authors comment on the availability of genetic testing for TRNT1 mutations and inheritance pattern.
  • Other biochemical abnormalities not mentioned in Table 1 have been reported and these should be included.
  • Could the authors comment on the management of these patients?
  • Can a distinction be made between paediatric and adult forms of the disease and could this be discussed?
  • The study of Wedatilake et al. described progressive cerebellar atrophy which was detected by brain MRI - this should to be discussed.
  • Italicize the TRNT1 gene

Author Response

Reviewer 1.

  1. Although I enjoyed reading the Introduction and Section 2 (mitochondria and disease), I felt these two sections were far too long and it was only until Section 3 that the review started to discuss the focus of the paper. Could the authors combine Sections 1 and 2 to provide a more succinct introduction or rearrange the content so that the focus of the paper is introduced to the reader earlier on.

We have reorganized the manuscript as suggested by the reviewer.

  1. Could the text in Figure 1 be increased in size as the font is too small to read.

We have increased size of the text and colour contract of the 5’CG to improve readibility of Figure 1 as requested.

  1. I would suggest including tRNA nucleotidyl transferase 1, tRNA CCA and TRNT1 as key words.

We have included these terms as key words as suggested.

  1. Please include the Enzyme Commission number for TRNT1.

We have included EC number for TRTN1 into the text of the manuscript.

  1. Are there any other names that TRNT1 is known by?

We are unaware of other names for TRNT1. In the past it was referred to as CCA1, however this is now the name of the equivalent enzyme within yeast.

  1. Could the authors please provide information on the chromosome location of TRNT1, the number of exons and span size.

This information is now included in the revised manuscript.

  1. Could the authors comment on the availability of genetic testing for TRNT1 mutations and inheritance pattern.

We have included this information in the Conclusion section of the revised manuscript as requested.

  1. Other biochemical abnormalities not mentioned in Table 1 have been reported and these should be included.

We have made an effort to include all relevant information for the available cases. Given the wide range of symptomps and clinical and biochemical features we appologize if we have missed something. We are aware that we have not included the detailed histology from the Barton (2018) study, and the severe detmatitis of one patient from Frans (2017) study. This was done for consistency reasons within the table. However, we provide references for each study where these and additional detaisl can be found.

  1. Could the authors comment on the management of these patients?

We hae included comment on the patient management in the conclusoin section of the revised manuscript.

  1. Can a distinction be made between paediatric and adult forms of the disease and could this be discussed?

Unfortunately this is not possible, given the short life span of majority of the patients. We have included comment on this in the revised manuscript.

  1. The study of Wedatilake et al. described progressive cerebellar atrophy which was detected by brain MRI - this should to be discussed.

We have included this information in the revised manuscript as requested.

  1. Italicize the TRNT1 gene.

All occurences of TRNT1 gene were italicised (excluding references).

Reviewer 2 Report

The review by Slade and Kattini et al. on human genetic disorders linked to TRNT1 mutations is well written and informative. It is always striking to learn how many different disease mutations have been already identified for a single gene carrying out a quite specific small task in cells. I have only a few suggestions.

It would be interesting to know more about the prevalence of the various mutations in human populations. How much is known or are the described mutations seen only in isolated families? How about the ethnic background, any evidence for bottlenecks or enrichment due to consanguineous marriages?

Minor notes on the text:

L28: [...] limited yet has great potential [...]

Awkward, Please modify.

L37: Check the sentence structure.

L39: “A common nuclear cause […] Nuclear cause? Please modify. How common are protein synthesis defects vs. e.g. mtDNA maintenance disorders? Examples of such diseases, references?

L44: Point mutation accumulation? Are there examples of such diseases?

L46: “mt DNA” ->mtDNA

L55: “to homeostasis”? What homeostasis?

Table 1 is nice and informative. Population prevalence data could be included here.

L284: Strange symbol with the IFN?

Figure 2 and 3 text too small

Abbreviations: “B_cell” ->B-cell

Author Response

Reviewer 2.

  1. It would be interesting to know more about the prevalence of the various mutations in human populations. How much is known or are the described mutations seen only in isolated families? How about the ethnic background, any evidence for bottlenecks or enrichment due to consanguineous marriages?

Given the rare nature of this disease and reported TRNT1 mutations there is insufficient information on this. We have provided a comment addressing this issue in the revised manuscript.

  1. L28: [...] limited yet has great potential [...] Awkward, Please modify.

We have changed this to ....'limited, yet has the potential to greatly improve.....'

  1. L37: Check the sentence structure.

We have verified the sentence but the structure seems OK to us.

  1. L39: “A common nuclear cause […] Nuclear cause? Please modify. How common are protein synthesis defects vs. e.g. mtDNA maintenance disorders? Examples of such diseases, references?

We have change the sentence to better reflect the meaning. (A common nuclear-encoded cause….). The references are provided.

  1. L44: Point mutation accumulation? Are there examples of such diseases?

We modified the sentence for clarify (point mutations). Leber’s hereditary optic neuropathy would be one such example. The reference is provided for readers who wish to pursue this further.

  1. L46: “mt DNA” ->mtDNA

Corrected as requested.

  1. L55: “to homeostasis”? What homeostasis?

The sentence was modified to read: ‘....homeostasis of ions ‘

  1. Table 1 is nice and informative. Population prevalence data could be included here.

Unfortunately that data is not available, given the scarcity of TRTN1 mutations in the population. Please see also the response to point 1.

  1. L284: Strange symbol with the IFN?

This was corrected – IFNalpha.

  1. Figure 2 and 3 text too small.

We have revised Figure 2 and 3 and increase the font size to improve readability.

  1. Abbreviations: “B_cell” ->B-cell.

This was corrected as requested.